# Dual-MoE: Learning Time and Channel Dependencies via Dual Mixture-of-Experts for Time Series Forecasting

## Abstract

Multivariate time series forecasting holds significant value in finance, energy, and transportation systems, yet faces critical challenges in jointly modeling temporal heterogeneity and dynamic channel dependencies. Existing approaches exhibit limitations in balancing long-term trends with short-term fluctuations, while struggling to capture time-varying inter-variable relationships. This paper proposes Dual-MoE, a dual mixture-of-experts framework that synergistically integrates temporal and channel modeling. The temporal expert dynamically combines multi-scale historical features (e.g., hourly details and weekly patterns) through adaptive gating mechanisms, whereas the channel expert learns dependency weights between variables via frequency-aware interaction modeling. Extensive experiments on real-world datasets demonstrate Dual-MoE's superior forecasting accuracy and robustness compared to state-of-the-art baselines. Its modular architecture provides a flexible and scalable paradigm for complex temporal dependency modeling, paving the way for further advancements in time series analysis. Code is available in Appendix.

## 1 Introduction

Multivariate Time Series Forecasting (MTSF) plays a crucial role in domains such as financial analysis, healthcare, electrical grid management, and environmental monitoring (Huang et al., 2024; Yang et al., 2023; Wang et al., 2024; Wu et al., 2024). Accurate long-term forecasting is essential for strategic decision-making and resource optimization. However, MTSF often requires modeling both temporal and inter-variable (e.g., channel) dependencies, which introduces two key challenges: *Temporal Distribution Shift* and *Noisy Channel Dependencies*. These challenges not only complicate model design but also significantly affect prediction stability and generalization performance across different real-world scenarios.

**Temporal Distribution Shift** stems from the non-stationary nature of data evolution (Kim et al., 2021; Liu et al., 2022b), making it difficult to predict future values using a fixed-length lookback window. In relatively stable periods, long-term historical patterns are beneficial, whereas during abrupt changes, recent short-term trends become more critical. Fixed-length windows struggle to balance these influences, failing to disentangle long- and short-term patterns. This highlights the need for a temporal fusion mechanism that dynamically integrates both patterns.

**Noisy Channel Dependencies** refer to dynamic inter-variable relationships in multivariate time series, where some correlations are meaningful and others are noisy or irrelevant. Traditional methods either assume full dependence (Liu et al., 2024a; Zhou et al., 2021; Zhang & Yan, 2023) or complete independence (Zeng et al., 2023; Nie et al., 2023), ignoring partial and time-varying dependencies. Prior approaches typically adopt static modeling, which lacks flexibility. An adaptive mechanism is needed to selectively enhance informative relationships while suppressing noise.

Moreover, time series often contain noise, outliers, or abrupt changes, which are inherently unpredictable and can degrade model performance (Xu et al., 2024; Xue et al., 2024). These elements cannot be captured by limited data features and may lead to overfitting if forcefully fitted. Hence, identifying and attenuating such noise is vital to improve robustness.

To address these challenges, we propose the **Dual-MoE** framework, which consists of three key modules: *Temporal Fusion MoE*, *Channel Fusion MoE*, and *Mask Loss Function*. (1) The Temporal Fusion MoE formulates the fusion of long- and short-term temporal patterns as a classification problem, leveraging Exponential Moving Average (EMA) to track trends and distributional shifts. It dynamically adjusts the influence of each component, ensuring that the model captures both transient and persistent dynamics. (2) The Channel Fusion MoE models noisy inter-variable dependencies via a learnable probability matrix that quantifies cross-channel relevance and adaptively fuses information across channels. This matrix acts as a flexible controller that emphasizes important relationships while filtering out interference. (3) Finally, to mitigate the effects of noise and outliers, we introduce a quantile-based Mask Loss Function, which dynamically masks unreliable loss components and enables the model to focus on predictable patterns, thereby improving both training stability and inference accuracy.

Overall, the Dual-MoE framework provides a unified and modular solution to address temporal shifts, dynamic channel dependencies, and noisy observations. Our experiments across a variety of datasets demonstrate that Dual-MoE consistently improves forecasting accuracy, particularly on smaller or lower-quality datasets where noise and instability are more pronounced. In addition, we provide theoretical insights into how lookback window length, data quality, and model granularity interact, offering new perspectives for the future design of robust MTSF models.

- **Dual-MoE Framework.** We introduce a general MTSF framework with dual mixture-of-experts, Dual-MoE, over temporal and channel dimensions for accurate and adaptive forecasting.

- **Empirical Validation.** Experiments across diverse benchmarks confirm the significant benefits of temporal and channel fusions at inference time, especially under complex distributional dynamics.

- **Experimental Insights.** We analyze the trade-off between long-term pattern capture and noise accumulation, revealing the role of lookback window length in TSF accuracy, generalization, and robustness.

## 2 RELATED WORKS

### 2.1 TIME SERIES FORECASTING

Long-term time series forecasting (LTSF) aims to predict future values over extended horizons based on historical observations. The prediction length denotes the number of future steps, while the lookback window refers to the past range used for forecasting. As time series applications grow across domains, the importance of LTSF continues to rise.

To improve accuracy and efficiency, various architectures have been explored, including MLPs (Zeng et al., 2023; Li et al., 2023), Transformers (Nie et al., 2023; Liu et al., 2024a; Zhang & Yan, 2023), CNNs (Eldele et al., 2024; Liu et al., 2022a), and RNNs (Sutskever, 2014; Wu et al., 2024). Recent models such as PatchTST (Nie et al., 2023) leverage patch embeddings, while iTransformer (Liu et al., 2024a) and ModernTCN (Luo & Wang, 2024) combine attention and convolution to capture complex dependencies in high-dimensional time series.

### 2.2 LOOKBACK WINDOW

Early works often adopted short lookback windows to reduce cost, as seen in Informer (Zhou et al., 2021), PatchTST (Nie et al., 2023), and Timer-XL (Liu et al., 2024b), but this limited access to long-term context and led to oversmoothing. Later studies explored longer windows, revealing that each dataset has an optimal window size (Shi et al., 2024), suggesting fixed lengths may not generalize well.

Fixed windows impose an inductive bias by underutilizing distant historical patterns. Our findings show that long-range dependencies often provide valuable signals. To address this, we propose a temporal fusion strategy that adaptively combines short- and long-term patterns based on data characteristics, enhancing model robustness and forecasting accuracy.

## 3 COMPARING MODEL BEHAVIORS WITH VARIED LOOKBACK WINDOWS

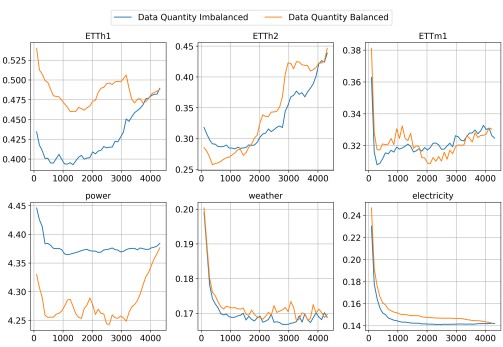

Figure 1: MSE trends for six datasets under different lookback window sizes.

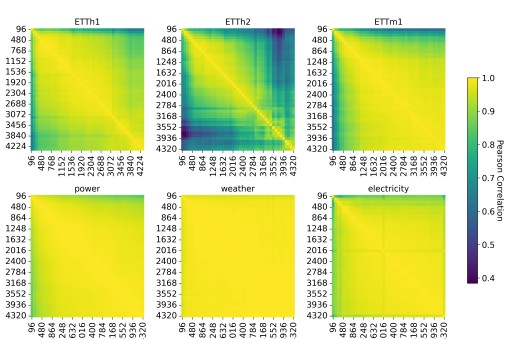

Figure 2: Loss Consistency heatmaps across varying lookback windows.

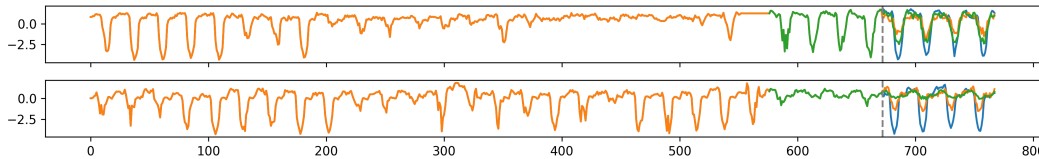

Figure 3: Illustrative examples showing that models with similar MSE can adopt distinct forecasting strategies.

To explore how lookback window sizes affect forecasting, we evaluate a Linear model on six datasets with lookback sizes ranging from 96 to 4380 (step=96). As shown in Figure 1, MSE typically follows a U-shaped trend: small windows lack context, medium windows improve performance, and overly long windows degrade accuracy due to noise or irrelevant patterns. Optimal window sizes differ by dataset—for instance, ETTh1 and ETTh2 favor short windows, while Electricity benefits from longer ones.

We also compare two data sampling strategies (see Appendix D.1): *Balanced* sampling, which ensures consistent sample size across lookback windows, and *Unbalanced* sampling, which does not restrict training data quantity. Interestingly, balanced sampling sometimes outperforms unbalanced even with fewer samples, indicating better temporal consistency and reduced distribution shift.

Figure 3 presents two example cases where different lookback windows lead to distinct decision strategies. In the first trial (top), short windows better capture local fluctuations; in the second (bottom), long windows better preserve global trends. Despite similar MSE, the prediction mechanisms differ significantly, highlighting that average error alone may obscure behavioral differences.

To quantify behavioral consistency, we propose **Loss Consistency**, which evaluates correlation between forecasting errors under different lookback settings:

$$\text{Loss Consistency} = \text{Corr}(\text{Loss}_i, \text{Loss}_k).$$

(Full derivation in Appendix D.2.)

As shown in Figure 2, datasets such as Weather show high consistency across window sizes, while ETTh2 and Power display sharp transitions, suggesting different internal decision logics. These patterns emphasize that performance stability and behavioral consistency must both be considered when choosing lookback configurations.

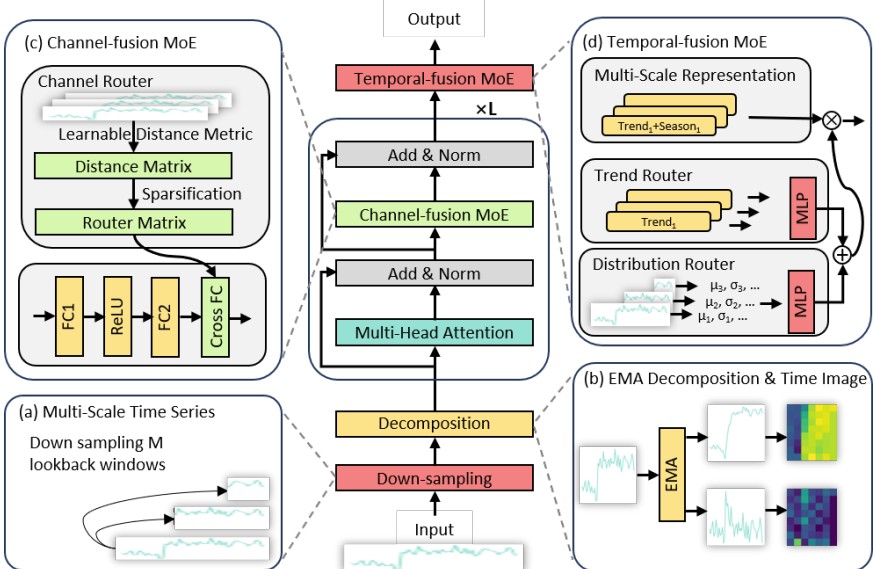

Figure 4: Illustration of the proposed dual-MoE framework, which integrates multi-stage feature extraction with both a Channel-fusion MoE and a Temporal-fusion MoE.

# 4 METHODOLOGY

## 4.1 OVERVIEW

Figure 4 presents the innovative architecture of Dual-MoE, enabling multi-scale temporal pattern modeling and cross-variable collaborative reasoning through multi-stage feature extraction and dynamic expert allocation. The Temporal-fusion MoE integrates dual routing mechanisms: Trend routers capture macroscopic evolutionary patterns using adaptive exponential smoothing, while distribution routers extract latent distribution, with dynamic weighted to fuse long-term and short-term temporal pattern. The Channel-fusion MoE module constructs variable dependency graphs in the frequency domain, implements differentiable Top-K expert selection based on mutual information maximization criteria, and enforces temporal dependencies through causal dilated convolution constraints. Final predictions are generated via residual gating mechanisms that fuse multi-level features followed by linear projection.

## 4.2 MULTI-SCALE LOOKBACK TIME IMAGING

**Exponential Decomposition.** Each univariate time series is decomposed using Exponential Moving Average (EMA), which emphasizes recent changes and better captures evolving trends. The resulting trend and seasonal components are separately encoded by Transformer blocks and fused via the dual-MoE framework.

Multi-scale Lookback Time Imaging starts from the longest lookback window $L_{max}$, generating multi-granularity time series segments through sliding window segmentation to directly capture local patterns at different time scales (e.g., fine-grained (micro) fluctuations, coarse-grained (macro) trends). Each lookback window corresponds to segments that are converted into time images via 1D-to-2D reshaping, mapping temporal neighborhood relationships to spatial adjacency in images (e.g., reshaping a window L 24 into a 4×6 image where adjacent time points correspond to neighboring pixels). The resulting 2D representations, with dimensions, are fed into the model in parallel. Cross-scale feature fusion enables simultaneous modeling of short-term details and long-term dependencies, particularly suited for scenarios with frequent abrupt changes or non-periodic time series.

**Transformer Encoder.** We adopt the standard Transformer encoder architecture (Liu et al., 2024a) with multi-head attention and position-wise feed-forward layers to extract latent representations from the time imaging inputs.

### 4.3 CHANNEL FUSION MOE

To enhance the interaction between channels while maintaining channel specialization, we introduce a channel fusion MoE Module. Each channel serves as an expert processor, and dynamic channel fusion is achieved by calculating a metric matrix and using a probability matrix to weight the channels.

**Expert Formalization**. We capture the similarity between channels by calculating the metric matrix. Specifically, for channel $n$, its expert embedding $E_n \in \mathbb{R}$ is computed as follows:

$$E_n = ReLU(W \cdot X_{n,:}^{chan} + b) \tag{1}$$

where $W \in \mathbb{R}^{D \times T}$ and $b \in \mathbb{R}^D$ are learnable parameters. The channel's representation is used to generate a $D$-dimensional expert embedding.

**Channel Distance Calculation**. We calculate the learnable Mahalanobis distance between each pair of channels, capturing the similarity between channels by computing their correlation. This metric matrix reflects the similarity between channels.

**Probability Matrix**. We introduce a probability matrix $P$, which dynamically weights the channels. This means that the probability value $P_{ij}$ for each channel represents the degree of correlation between channel $i$ and channel $j$. The higher the probability, the stronger the relationship between the channels. The probability matrix $P$ is computed by evaluating the relationships between channels:

$$P_{ij} = \begin{cases} 1 & \text{if } i = j, \\ 0 & \text{if } i \neq j \text{ and } \frac{\frac{\gamma}{D_{ik}}}{\max\left(\frac{1}{D_{ik}}\right)} < \delta, \\ \frac{\frac{\gamma}{D_{ik}}}{\max\left(\frac{1}{D_{ik}}\right)} & \text{if } i \neq j \text{ and } \frac{\frac{\gamma}{D_{ik}}}{\max\left(\frac{1}{D_{ik}}\right)} \geq \delta. \end{cases} \tag{2}$$

where $\gamma$ is a discount parameter that avoids full channel fusion and $\delta$ is the threshold to filter out noisy dependency.

**Channel Fusion**. The final prediction is obtained by weighted fusion of the outputs from all channels. Specifically, the final prediction $\hat{X}i$ for channel $i$ is obtained by weighting the outputs of all expert channels as follows:

$$\hat{X}i = \sum_{j=1}^{N} Pij \cdot FFN(X_j) \tag{3}$$

where $FFN$ is the expert network for channel $j$. The weight matrix $P$ is used to weight the contribution of each channel based on their similarity, preserving the contributions from channels with higher correlations and reducing the influence of less relevant channels.

### 4.4 TEMPORAL FUSION MOE

To comprehensively capture temporal patterns across diverse horizons, we propose a Temporal Fusion MoE module that dynamically aggregates predictions from multiple scale-specific encoders. The architecture operates through three coordinated phases:

**Statistical Representation:** For each univariate series $X_{n,:} \in \mathbb{R}^T$, we characterize its temporal dynamics via:

$$\mu_n = \frac{1}{T} \sum_{t=1}^{T} X_{n,t}, \quad \sigma_n = \sqrt{\frac{1}{T} \sum_{t=1}^{T} (X_{n,t} - \mu_n)^2} \tag{4}$$

$$\max_n = \max(X_{n,1:T}), \quad \min_n = \min(X_{n,1:T}) \tag{5}$$

These statistics form a descriptor vector $S_n \in \mathbb{R}^4$ that encodes amplitude, dispersion, and extremal characteristics of the time series.

**Distribution Router:** A Multi-Layer Perceptron (MLP) is applied to the statistical representation $S_n$ of different lookback windows to generate a set of weights for each window. The output of the MLP for each window is used to capture the significance of each lookback window in modeling the temporal dynamics of the series:

$$W_{\text{dist},n} = \text{MLP}(S_n) \tag{6}$$

where $W_{\text{dist},n}$ is the learned fusion weights for each window.

**Trend Router:** To capture the long-term trend dynamics, the final layer of the trend representation is input to another MLP, which generates weights that reflect the importance of the trend component for the prediction:

$$W_{\text{trend},n} = \text{MLP}(\text{TrendRep}_n) \tag{7}$$

where $\text{TrendRep}_n$ is the extracted trend representation for the time series, and $W_{\text{trend},n}$ are the weights of the trend component.

**Weight Fusion:** The final weight for each time step is obtained by summing the weights from the distribution and the trend routers:

$$W_n = W_{\text{dist},n} + W_{\text{trend},n} \tag{8}$$

$W_n$ is the importance of each lookback window in capturing both short-term distributional features and long-term trend patterns.

**Temporal Fusion:** The features extracted from different lookback windows are first flattened, then passed through a linear projection, and finally multiplied by their corresponding weights. The weighted features are summed across all lookback windows to obtain the final prediction:

$$\hat{X}_i = \sum_{j=1}^{N} W_{ij} \cdot \text{Linear}\left(\text{Flatten}(X_j)\right) \tag{9}$$

where $W_{ij}$ is the fusion weight for window $j$ at time step $i$, and $\text{Flatten}(X_j)$ is the flattened feature vector from window $j$. The output is then passed through a linear projection layer to map the aggregated features to the forecast horizon.

### 4.5 MASK LOSS FUNCTION

To enhance robustness against noise and unpredictable patterns, we design a quantile-based **Mask Loss Function** that selectively emphasizes more reliable channels during training.

Given prediction $\hat{Y} \in \mathbb{R}^{B \times T \times C}$ and ground truth $Y \in \mathbb{R}^{B \times T \times C}$, the element-wise squared error is:

$$\mathcal{L}_{b,t,c} = (\hat{Y}_{b,t,c} - Y_{b,t,c})^2 \tag{10}$$

$$\bar{\mathcal{L}}_c = \frac{1}{B \cdot T} \sum_{b=1}^{B} \sum_{t=1}^{T} \mathcal{L}_{b,t,c} \tag{11}$$

Let $\tau_q$ be the $q$-th quantile of $\{\bar{\mathcal{L}}_c\}_{c=1}^{C}$. The channel mask $m_c$ is:

$$m_c = \begin{cases} 1, & \bar{\mathcal{L}}_c \leq \tau_q \\ 0, & \text{otherwise} \end{cases} \tag{12}$$

The binary mask $m_c$ is then broadcasted to the full loss tensor:

$$M_{b,t,c} = m_c \in \{0, 1\} \tag{13}$$

$$\mathcal{L}_{\text{final}} = \frac{\sum_{b,t,c} \mathcal{L}_{b,t,c} \cdot M_{b,t,c}}{\sum_{b,t,c} M_{b,t,c} + \epsilon} \tag{14}$$

This approach allows the model to focus on predictable channels while suppressing the influence of noisy or highly volatile ones.

| Models | Dual-MoE | | xPatch | | PDF | | FITS | | iTransformer | | PatchTST) | | MICN | | DLinear | |
|---|---|---|---|---|---|---|---|---|---|---|---|---|---|---|---|---|
| Metric | MSE | MAE | MSE | MAE | MSE | MAE | MSE | MAE | MSE | MAE | MSE | MAE | MSE | MAE | MSE | MAE |
| ETTh1 | **0.389** | **0.418** | 0.413 | 0.423 | 0.407 | 0.426 | 0.408 | 0.427 | 0.439 | 0.448 | 0.419 | 0.436 | 0.420 | 0.447 | 0.425 | 0.439 |
| ETTh2 | 0.337 | **0.379** | 0.341 | 0.383 | 0.347 | 0.391 | **0.335** | 0.386 | 0.370 | 0.403 | 0.351 | 0.395 | 0.482 | 0.472 | 0.470 | 0.468 |
| ETTm1 | **0.340** | **0.367** | 0.351 | 0.371 | 0.342 | 0.376 | 0.357 | 0.377 | 0.361 | 0.390 | 0.349 | 0.381 | 0.355 | 0.383 | 0.356 | 0.378 |
| ETTm2 | **0.247** | **0.303** | 0.251 | 0.307 | 0.250 | 0.313 | 0.254 | 0.313 | 0.269 | 0.327 | 0.256 | 0.314 | 0.294 | 0.357 | 0.259 | 0.324 |
| Weather | **0.217** | **0.247** | 0.219 | 0.249 | 0.227 | 0.263 | 0.244 | 0.281 | 0.232 | 0.270 | 0.224 | 0.261 | 0.239 | 0.289 | 0.242 | 0.293 |
| Traffic | 0.390 | 0.255 | **0.388** | **0.245** | 0.395 | 0.270 | 0.429 | 0.302 | 0.397 | 0.281 | 0.397 | 0.275 | 0.539 | 0.313 | 0.418 | 0.287 |
| Electricity | **0.153** | **0.244** | 0.154 | 0.245 | 0.160 | 0.253 | 0.169 | 0.265 | 0.163 | 0.258 | 0.171 | 0.270 | 0.179 | 0.290 | 0.167 | 0.264 |
| Exchange | 0.328 | **0.384** | 0.361 | 0.401 | 0.350 | 0.397 | 0.349 | 0.396 | 0.360 | 0.404 | 0.322 | 0.385 | 0.321 | 0.393 | **0.292** | 0.391 |
| Solar | **0.183** | 0.215 | 0.192 | **0.212** | 0.200 | 0.263 | 0.232 | 0.268 | 0.202 | 0.262 | 0.200 | 0.284 | 0.254 | 0.302 | 0.224 | 0.286 |
| ILI | **1.583** | 0.788 | 1.633 | **0.783** | 1.808 | 0.898 | 2.334 | 1.052 | 1.857 | 0.892 | 1.902 | 0.879 | 2.368 | 1.049 | 2.185 | 1.040 |
| $1^{st}$ Count | 14 | | 3 | | 0 | | 1 | | 0 | | 0 | | 0 | | 1 | |

Table 1: Fair long-term forecasting results under hyperparameter searching without the "drop-last" trick. Results are averaged (Avg) over all horizons for each dataset. The best model is **bold**, and the second best is underlined. Count is the number of the best results.

## 5 EXPERIMENTS

### 5.1 EXPERIMENTS SETTINGS

**Datasets** Following previous research (Qiu et al., 2024; Zhou et al., 2021; Jin et al., 2024; Nie et al., 2023), we conduct evaluations on ten well-acknowledged real-world datasets, spanning various applications, including Electricity Transformer Temperature (ETT) (Zhou et al., 2021); Traffic; Electricity; Weather; National Illness (ILI) (Lai et al., 2018); and Exchange. Each dataset consists of multivariate time series data. Detailed descriptions of each dataset's characteristics and how they are processed are available as follows and in Appendix.

**Baselines.** We choose the latest state-of-the-art models to serve as baselines, including CNN-based models (xPatch (Stitsyuk & Choi, 2025), MICN (Wang et al., 2023)), MLP-based models (FITS (Xu et al., 2024), and DLinear (Zeng et al., 2023)), and Transformer-based models (PDF (Dai et al., 2024), iTransformer (Liu et al., 2024a), PatchTST (Nie et al., 2023).

**Implementation Details.** We conduct all experiments on an NVIDIA GeForce RTX 3090 GPU with 64-bit Linux 5.15.0-56-generic, with 60/20/20 training/validation/testing split for ETTs and Solar, and 70/10/20 for other datasets.

### 5.2 MAIN RESULTS

Comprehensive forecasting results are listed in Table. 1 and Table. 8, with the best results in **bold** and the second-best in underlined. The key findings from our experiments are summarized as follows:

1) **Enhanced Predictive Accuracy:** Dual-MoE consistently outperforms diverse baseline models across various architectures, achieving 71 wins out of 100 tests under different metrics and settings. Specifically, among 40 forecasting tasks, it ranks first overall, surpassing the second-best model, xPatch, by ∼3.0% in average MSE and ∼0.3% in average MAE. In addition, our Dual MoE performs robustly in both low-dimensional datasets (e.g. ETTs wiith 7 dim) and high-dimensional ones (e.g. Electricity with 321 dim), demonstrating the Dual-MoE's strong predictive ability across a wide range of complex scenarios.

2) **Effectiveness in Handling Temporal Distribution Shifts:** A key strength of Dual-MoE lies in its ability to tackle temporal heterogeneity, particularly in the presence of temporal distribution shifts. Compared to Recent PDF model, our Dual-MoE achieves a notable reduction of ∼4.6% and ∼6.0% in average MSE and MAE. These results highlight the effectiveness of the Temporal Fusion MoE module in modeling dynamic, non-stationary temporal patterns, making it more adaptable to time series with evolving trends.

3) **Advantages of Channel Fusion Mechanism:** Dual-MoE effectively leverages its channel fusion mechanism to outperform channel-independent-based models, such as TimeMixer and PatchTST, particularly on large-scale datasets with strong inter-channel correlations (e.g., Solar, Traffic, and Electricity). In contrast, SOTA channel-dependent models like iTransformer and TimesNet under-perform on ETTs due to weak channel correlations, making them vulnerable to noise from unrelated channels and leading to performance degradation. Dual-MoE addresses this limitation through its

Channel Fusion MoE module, which dynamically identifies and attends to the most relevant channels, mitigating noise interference and enhancing performance on datasets with weak or varying inter-channel dependencies.

### 5.3 MODEL ANALYSIS

#### 5.3.1 ABLATION STUDY.

To ascertain the impact of each module in Dual-MOE, we perform ablation studies. (1) *w/o T-MoE:* Remove the Temporal fusion Mixture-of-Experts Module. (2) *w/o C-MoE:* Remove the Channel fusion Mixture-of-Experts Module. (3) *w/o EMA:* Remove the Exponential Moving Average Module. (4) *w/o Mask Loss:* Remove the Mask Loss function Module.

Table 2 reflects the results of each component in Dual-MOE, as removing any module may degrade performance. On the Weather dataset, where variables exhibit strong inherent correlations, removing the C-MoE causes the most significant drop, emphasizing its pivotal role in coordinating cross-channel interactions. Conversely, ETTh2, which exhibits non-stationary trends and distribution shifts, is particularly sensitive to the removal of the T-MoE, reflecting its importance in adapting to evolving temporal patterns. The absence of either the EMA or Mask Loss also leads to observable declines, validating their contributions to training stability and feature robustness. These dataset-specific sensitivities demonstrate the complementary nature of each module and their collective contribution to Dual-MoE's ability to model temporal dynamics, channel dependencies, and training challenges within a unified framework.

| Variants | ETTh2 (avg.) | | Weather (avg.) | |
|---|---|---|---|---|
| | MSE | MAE | MSE | MAE |
| w/o T-MoE | 0.346 | 0.391 | 0.223 | 0.255 |
| w/o C-MoE | 0.343 | 0.393 | 0.232 | 0.261 |
| w/o EMA | 0.341 | 0.386 | 0.219 | 0.250 |
| w/o Mask Loss | 0.342 | 0.385 | 0.221 | 0.252 |
| Dual-MoE | **0.337** | **0.380** | **0.217** | **0.247** |

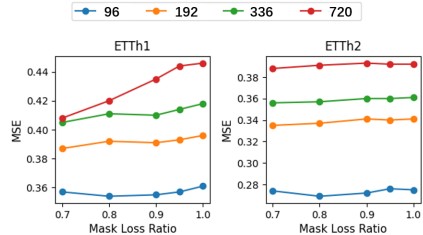

Table 2: Ablation study. Results are averaged from all forecasting horizons $\in \{96, 192, 336, 720\}$

Table 3: MSE across different mask loss ratios on ETTh1 and ETTh2. Each line is a forecast horizon.

#### 5.3.2 COMPARISON AMONG DIFFERENT LOSS MASK RATIOS.

To evaluate the effectiveness of the Mask Loss function, we evaluate those models under varying mask loss ratios, ranging from 0.7 to 1. As shown in Figure 3, MSE values on ETTh1 and ETTh2 are plotted across various forecasting horizons. The results clearly demonstrate that the mask loss function significantly contributes to training stability and model performance. For both datasets, MSE varies with the mask loss ratio, indicating that proper tuning is essential for optimal results. From the figure, we can observe that for the ETTh1 dataset, higher mask loss ratios consistently lead to better performance, while ETTh2 shows a subtler but consistent trend. This suggests that appropriate tuning of the mask loss ratio is essential for robust performance, particularly in datasets with complex temporal dynamics. Moreover, the effect is more pronounced at longer horizons, highlighting the importance of Mask Loss in long-term forecasting.

#### 5.3.3 PARAMETER SENSITIVITY: VARYING THE NUMBER OF EXPERTS.

The Dual-MoE's Temporal Fusion module dynamically adjusts to different historical contexts by varying both the number of experts $M$ and the lookback window sizes. We conduct a parameter sensitivity analysis using the following setups: ETTh1, ETTh2, Solar: Lookback windows $\{672, 512, 336, 96\}$, tested for $M \in \{1, 2, 3, 4\}$ experts. Exchange: Shorter windows $\{96, 72, 48, 24\}$, tested for $M \in \{1, 2, 3, 4\}$ experts. Table 4 summarizes the impact of different $M$ values on forecasting accuracy, revealing three key insights:

| Windows | M=1 | | M=2 | | M=3 | | M=4 | |
|---------|-----|-----|-----|-----|-----|-----|-----|-----|
| Metrics | MSE | MAE | MSE | MAE | MSE | MAE | MSE | MAE |
| ETTh1 | 0.394 | 0.420 | 0.393 | 0.422 | **0.389** | **0.418** | 0.396 | 0.425 |
| ETTh2 | 0.346 | 0.391 | 0.342 | 0.387 | **0.337** | **0.380** | 0.345 | 0.389 |
| Exchange | 0.385 | 0.405 | 0.350 | 0.399 | 0.334 | 0.388 | **0.328** | **0.384** |
| Solar | 0.185 | 0.218 | **0.183** | **0.215** | 0.193 | 0.242 | 0.195 | 0.253 |

Table 4: Parameter sensitivity study. The prediction accuracy varies with M, which is the number of lookback windows. Results are averaged from all forecasting horizons.

(1) **Universal Improvement**: Single-expert models ($M = 1$) consistently underperform, confirming the advantage of multiple specialized experts.

(2) **Domain-Specific Patterns**: Datasets from related domains exhibit similar optimal expert counts. For instance, ETTh1 and ETTh2 achieve peak performance at M=3, suggesting shared temporal characteristics within the energy domain.

(3) **Cross-Domain Variation**: Optimal $M$ varies by domain—Solar (energy) performs best with M=2, while Exchange (financial) requires M=4, reflecting fundamental differences in temporal dynamics and channel complexity.

### 5.3.4 EFFICIENCY ANALYSIS.

As shown in Table 5, Dual-MoE achieves superior accuracy **with only 48% MACs, and 18% Inference Time** compared to Pathformer. These results indicate that our dual-routing design not only provides accuracy gains but also achieves high computational efficiency on large-scale benchmarks.

| Method | MACs | Infer. Time | MSE |
|--------|------|-------------|-----|
| Pathformer | 8.69G | 156.94ms | 0.211 |
| PDF | 7.76G | 58.78ms | 0.199 |
| Dual-MoE | 3.77G | 28.77ms | 0.185 |

| Dataset | Entropy | Stationarity | Impr. |
|---------|---------|--------------|-------|
| Exchange | 0.805 | -1.902 | **+9.1%** |
| ETTh1 | 0.775 | -5.909 | **+5.8%** |
| Electricity | 0.516 | -8.445 | **+0.6%** |

Table 5: Number of MACs, inference time, and MSE of TSF models under look-back window=512 and forecasting horizon=720 on the large Electricity dataset.

Table 6: Comparison across datasets of different quality levels. MSE improvement is relative to xPatch.

### 5.3.5 DATASET QUALITY AND PERFORMANCE.

To better understand the model's effectiveness under different data characteristics, we analyze three datasets with varying sequence lengths, entropy, and stationarity levels. As shown in Table 6, Dual-MoE achieves the most substantial improvements over xPatch on datasets with higher entropy and lower stationarity (e.g., Exchange with +9.1% MSE gain), while performance gains diminish on more stable and regular datasets like Electricity (+0.6%).

## 6 CONCLUSION

In this paper, we propose Dual-MoE, a novel framework for multivariate time series forecasting that synergizes temporal and channel-wise dynamic modeling. It integrates a Temporal-fusion MoE which introduces dual routing mechanisms (trend-aware and distribution-sensitive routers) to adaptively aggregate multi-scale temporal patterns, effectively disentangling long-term evolutionary trends and short-term fluctuations through exponential decomposition. Second, the Channel-fusion MoE employs frequency-domain metric learning with sparse differentiable routing, enabling interpretable cross-channel collaboration while preserving specialized representations via orthogonal expert constraints and load-balanced activation. Extensive experiments validate the superiority of Dual-MoE in handling complex temporal dependencies.

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

## A  DATASET DESCRIPTIONS

Table 7: Dataset detailed descriptions. *Dim* denotes the number of variables per dataset, i.e., channels. *Frequency* represents the sampling interval of time points.

| Dataset | Dim | Timesteps | Frequency | Domain |
|---------|-----|-----------|-----------|--------|
| ETTh1&h2 | 7 | 17420 | Hourly | Electricity |
| ETTm1&m2 | 7 | 69,680 | 15-min | Electricity |
| Exchange | 8 | 7588 | Daily | Economy |
| Electricity | 321 | 26304 | Hourly | Electricity |
| Traffic | 862 | 17544 | Hourly | Transportation |
| Weather | 21 | 52696 | 10-min | Weather |
| ILI | 7 | 966 | Weakly | Disease |
| Solar | 137 | 52560 | 10-min | Energy |

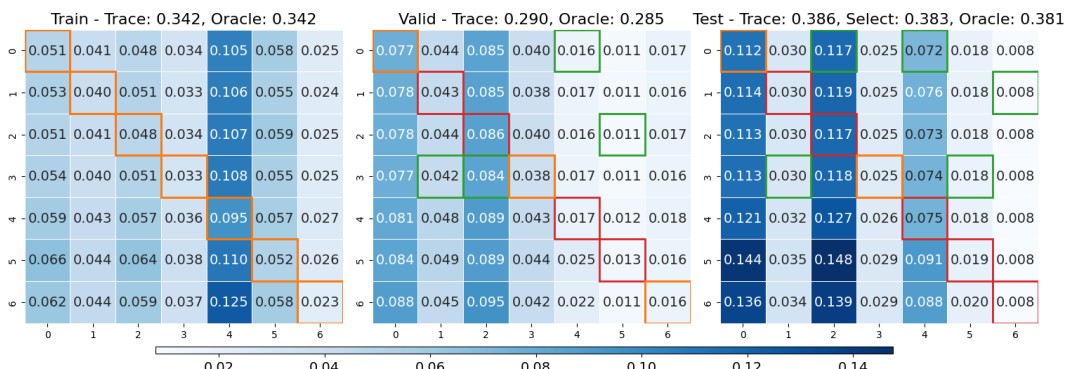

Figure 5: The results of cross-channel forecasting experiments.

- **ETT** consists of datasets with varying granularities, including two hourly-level datasets (ETTh1, ETTh2) and two 15-minute-level datasets (ETTm1, ETTm2). These datasets feature six power load variables and the target variable "oil temperature," spanning from July 2016 to July 2018.

- **Traffic** tracks hourly road occupancy on San Francisco freeways over the period of 2015–2016.

- **Electricity** provides hourly power usage data from 321 customers, collected from 2012 to 2014.

- **Exchange-Rate** contains daily foreign exchange rate data for eight countries, with records from 1990 to 2016.

- **Weather** consists of 21 weather parameters, including air temperature and humidity, recorded at 10-minute intervals throughout 2020 in Germany.

- **ILI** is sourced from the U.S. Centers for Disease Control and Prevention (CDC), documenting weekly instances of influenza-like illness from 2002 to 2021, including patient counts and ratios.

- **Solar** provides solar energy production data from 137 photovoltaic (PV) plants in Alabama.

**Drop Last Issue.** Several studies (Xu et al., 2024; Qiu et al., 2024) have highlighted the complications of using the "drop-last" setting during model evaluation. Specifically, enabling " drop_last=True" can lead to errors because of changes in the batch size of the test set. To mitigate these issues, we intentionally set the "drop_last=False" option for our experiments.

## B    CROSS CHANNEL FORECASTING

To further investigate the channel dependencies, we train models on one channel and test them on another using the ETTh1 dataset, as shown in Figure 5. The results are presented in a matrix format where the rows represent the training channels and the columns represent the testing channels, with separate matrices for training, validation, and testing performances. Our findings reveal consistent results across columns, but substantial variability across rows, highlighting the significant impact of the testing channel on performance. The diagonal entries of the matrix (trace), marked in yellow or red, correspond to models trained and tested on the same channel, while green entries indicate improved performance from cross-channel training. Notably, when training on channels 0 to 3, the model's performance significantly improves. While training on channels 5 to 6, the model's performance decreases. This underscores the strong inter-channel dependencies and the disadvantages of training on noisy channels.

## C MODULE DETAILS

### C.1 EXPONENTIAL DECOMPOSITION

Employing the Exponential Moving Average (EMA) method, each univariate time series is decomposed into trend and seasonal components. EMA is a more flexible smoothing technique than traditional moving averages because it assigns exponentially decreasing weights to past observations, with more recent data points having greater importance. This allows the model to better capture underlying trends and adapt more quickly to abrupt changes compared to the traditional moving average method, which applies equal weights to all past data points, potentially leading to slower adaptation in the presence of sudden shifts. These components undergo independent feature learning through an identical Transformer architecture. The extracted trend and seasonal features are then fused and processed via a dual-MoE framework incorporating lookback window integration to generate final predictions.

### C.2 TRANSFORMER ENCODER

We use a vanilla Transformer encoder that maps the time imaging to the latent representations. The encoder processes time imaging $\mathbf{X}_p \in \mathbb{R}^{N \times P}$ through linear projection $\mathbf{W}_e \in \mathbb{R}^{P \times D}$ and position encoding $\mathbf{E}_p \in \mathbb{R}^{N \times D}$, generating initial embeddings $\mathbf{H}_0 = \mathbf{X}_p \mathbf{W}_e + \mathbf{E}_p$. Each encoder layer computes multi-head attention through $H$ parallel heads, where each head applies scaled dot-product attention:

$$\text{head}_h = \text{Softmax}\left(\frac{(\mathbf{H}\mathbf{W}_h^Q)(\mathbf{H}\mathbf{W}_h^K)^\top}{\sqrt{d_k}}\right)\mathbf{H}\mathbf{W}_h^V$$

with projection matrices $\mathbf{W}_h^Q, \mathbf{W}_h^K \in \mathbb{R}^{D \times d_k}$ and $\mathbf{W}_h^V \in \mathbb{R}^{D \times d_v}$. Concatenated outputs pass through batch normalization before residual summation.

The position-wise feed-forward network expands features to $4D$ dimensions using GELU activation, implemented as $\text{FFN}(\mathbf{H}) = \text{GELU}(\mathbf{H}\mathbf{W}_1)\mathbf{W}_2$ with $\mathbf{W}_1 \in \mathbb{R}^{D \times 4D}$ and $\mathbf{W}_2 \in \mathbb{R}^{4D \times D}$.

## D EXPERIMENTAL DETAILS

### D.1 D.1 BALANCED VS IMBALANCED

We use two sampling strategies for training with different lookback window sizes:

- **Balanced Sampling:** For each lookback window length, the number of training samples is fixed, ensuring consistent data scale across experiments. This helps isolate the effect of lookback length itself.
- **Unbalanced Sampling:** No restriction on the number of training samples. Longer lookback windows naturally yield fewer samples due to sequence overlap.

Balanced sampling filters out potentially distribution-shifted data and often leads to more stable performance, especially on datasets like Power and ETTh2.

### D.2 LOSS CONSISTENCY

Given ground truth sequence $y = \{y_1, \ldots, y_N\}$ and prediction $\hat{y}_i = \{\hat{y}_{i1}, \ldots, \hat{y}_{iN}\}$ from model $M_i$, we define the MSE loss for the $j$-th sample as:

$$\text{Loss}_{ij} = \frac{1}{T}\sum_{t=1}^{T}(y_{jt} - \hat{y}_{ijt})^2,$$

where $T$ is the prediction length.

Let $\text{Loss}_i = [\text{Loss}_{i1}, \ldots, \text{Loss}_{in}]$ and $\text{Loss}_k = [\text{Loss}_{k1}, \ldots, \text{Loss}_{kn}]$ be the loss vectors of models $M_i$ and $M_k$ over all trials. Then, we compute:

$$\text{Loss Consistency} = \text{Corr}(\text{Loss}_i, \text{Loss}_k),$$

where $\mathrm{Corr}(\cdot, \cdot)$ is Pearson correlation. A higher value indicates stronger agreement in error distributions across different lookback settings.

## D.3 Full Results

| Models | Metric | Dual-MoE (ours) | | xPatch (2025) | | PDF (2024) | | FITS (2024) | | iTransformer (2024) | | PatchTST (2023) | | MICN (2023) | | DLinear (2023) | |
|---|---|---|---|---|---|---|---|---|---|---|---|---|---|---|---|---|---|
| | | MSE | MAE | MSE | MAE | MSE | MAE | MSE | MAE | MSE | MAE | MSE | MAE | MSE | MAE | MSE | MAE |
| ETTh1 | 96 | **0.354** | **0.390** | 0.363 | 0.390 | 0.360 | 0.391 | 0.376 | 0.396 | 0.386 | 0.405 | 0.377 | 0.397 | 0.378 | 0.412 | 0.379 | 0.403 |
| | 192 | **0.388** | **0.413** | 0.404 | 0.414 | 0.392 | 0.414 | 0.400 | 0.418 | 0.424 | 0.440 | 0.409 | 0.425 | 0.400 | 0.430 | 0.408 | 0.419 |
| | 336 | **0.405** | **0.429** | 0.432 | 0.432 | 0.418 | 0.435 | 0.419 | 0.435 | 0.449 | 0.460 | 0.431 | 0.444 | 0.428 | 0.447 | 0.440 | 0.440 |
| | 720 | **0.408** | **0.438** | 0.451 | 0.457 | 0.456 | 0.462 | 0.435 | 0.458 | 0.495 | 0.487 | 0.457 | 0.477 | 0.474 | 0.499 | 0.471 | 0.493 |
| | Avg | **0.389** | **0.418** | 0.413 | 0.423 | 0.407 | 0.426 | 0.408 | 0.427 | 0.439 | 0.448 | 0.419 | 0.436 | 0.420 | 0.447 | 0.425 | 0.439 |
| ETTh2 | 96 | **0.269** | **0.330** | 0.274 | 0.333 | 0.276 | 0.341 | 0.277 | 0.345 | 0.297 | 0.348 | 0.274 | 0.337 | 0.313 | 0.372 | 0.300 | 0.364 |
| | 192 | 0.335 | 0.374 | 0.336 | 0.374 | 0.339 | 0.382 | **0.331** | 0.379 | 0.372 | 0.403 | 0.348 | 0.384 | 0.419 | 0.439 | 0.387 | 0.423 |
| | 336 | 0.356 | **0.390** | 0.366 | 0.400 | 0.374 | 0.406 | **0.350** | 0.396 | 0.388 | 0.417 | 0.377 | 0.416 | 0.474 | 0.475 | 0.490 | 0.487 |
| | 720 | **0.386** | **0.422** | 0.388 | 0.425 | 0.398 | 0.433 | 0.382 | 0.425 | 0.424 | 0.444 | 0.406 | 0.441 | 0.723 | 0.600 | 0.704 | 0.597 |
| | Avg | 0.337 | **0.379** | 0.341 | 0.383 | 0.347 | 0.391 | **0.335** | 0.386 | 0.370 | 0.403 | 0.351 | 0.395 | 0.482 | 0.472 | 0.470 | 0.468 |
| ETTm1 | 96 | **0.277** | **0.327** | 0.287 | 0.330 | 0.286 | 0.340 | 0.303 | 0.345 | 0.300 | 0.353 | 0.289 | 0.343 | 0.303 | 0.349 | 0.300 | 0.345 |
| | 192 | **0.324** | **0.355** | 0.328 | 0.356 | 0.321 | 0.364 | 0.337 | 0.365 | 0.341 | 0.380 | 0.329 | 0.368 | 0.336 | 0.369 | 0.336 | 0.366 |
| | 336 | 0.355 | **0.376** | 0.363 | 0.379 | **0.354** | 0.383 | 0.368 | 0.384 | 0.374 | 0.396 | 0.362 | 0.390 | 0.370 | 0.391 | 0.367 | 0.386 |
| | 720 | **0.404** | **0.408** | 0.426 | 0.417 | 0.408 | 0.415 | 0.420 | 0.413 | 0.429 | 0.430 | 0.416 | 0.423 | 0.410 | 0.421 | 0.419 | 0.416 |
| | Avg | **0.340** | **0.367** | 0.351 | 0.371 | 0.342 | 0.376 | 0.357 | 0.377 | 0.361 | 0.390 | 0.349 | 0.381 | 0.355 | 0.383 | 0.356 | 0.378 |
| ETTm2 | 96 | **0.157** | **0.242** | 0.157 | 0.243 | 0.163 | 0.251 | 0.165 | 0.254 | 0.175 | 0.266 | 0.165 | 0.255 | 0.173 | 0.271 | 0.164 | 0.255 |
| | 192 | **0.215** | **0.282** | 0.216 | 0.285 | 0.219 | 0.290 | 0.219 | 0.291 | 0.242 | 0.312 | 0.221 | 0.293 | 0.232 | 0.313 | 0.224 | 0.304 |
| | 336 | **0.267** | **0.318** | 0.271 | 0.323 | 0.269 | 0.330 | 0.272 | 0.326 | 0.282 | 0.337 | 0.276 | 0.327 | 0.303 | 0.367 | 0.277 | 0.337 |
| | 720 | **0.348** | **0.371** | 0.358 | 0.377 | 0.349 | 0.382 | 0.359 | 0.381 | 0.375 | 0.394 | 0.362 | 0.381 | 0.467 | 0.477 | 0.371 | 0.401 |
| | Avg | **0.247** | **0.303** | 0.251 | 0.307 | 0.250 | 0.313 | 0.254 | 0.313 | 0.269 | 0.327 | 0.256 | 0.314 | 0.294 | 0.357 | 0.259 | 0.324 |
| Weather | 96 | **0.143** | 0.185 | 0.144 | **0.184** | 0.147 | 0.196 | 0.172 | 0.225 | 0.157 | 0.207 | 0.149 | 0.196 | 0.172 | 0.232 | 0.170 | 0.230 |
| | 192 | **0.187** | **0.223** | 0.188 | 0.227 | 0.193 | 0.240 | 0.215 | 0.261 | 0.200 | 0.248 | 0.191 | 0.239 | 0.214 | 0.270 | 0.216 | 0.275 |
| | 336 | **0.234** | **0.264** | 0.236 | 0.266 | 0.245 | 0.280 | 0.261 | 0.295 | 0.252 | 0.287 | 0.242 | 0.279 | 0.259 | 0.309 | 0.258 | 0.307 |
| | 720 | **0.304** | **0.316** | 0.309 | 0.319 | 0.323 | 0.334 | 0.326 | 0.341 | 0.320 | 0.336 | 0.312 | 0.330 | 0.309 | 0.343 | 0.323 | 0.362 |
| | Avg | **0.217** | **0.247** | 0.219 | 0.249 | 0.227 | 0.263 | 0.244 | 0.281 | 0.232 | 0.270 | 0.224 | 0.261 | 0.239 | 0.289 | 0.242 | 0.293 |
| Traffic | 96 | 0.359 | 0.239 | 0.359 | **0.233** | 0.368 | 0.252 | 0.400 | 0.280 | 0.363 | 0.265 | 0.370 | 0.262 | 0.517 | 0.313 | 0.395 | 0.275 |
| | 192 | 0.378 | 0.248 | **0.375** | **0.239** | 0.382 | 0.261 | 0.412 | 0.288 | 0.384 | 0.273 | 0.386 | 0.269 | 0.526 | 0.302 | 0.407 | 0.280 |
| | 336 | **0.387** | 0.256 | 0.388 | **0.244** | 0.393 | 0.268 | 0.433 | 0.308 | 0.396 | 0.277 | 0.396 | 0.275 | 0.545 | 0.307 | 0.417 | 0.286 |
| | 720 | 0.434 | 0.278 | **0.429** | **0.264** | 0.438 | 0.297 | 0.478 | 0.339 | 0.445 | 0.308 | 0.435 | 0.295 | 0.569 | 0.328 | 0.454 | 0.308 |
| | Avg | 0.390 | 0.255 | **0.388** | **0.245** | 0.395 | 0.270 | 0.429 | 0.302 | 0.397 | 0.281 | 0.397 | 0.275 | 0.539 | 0.313 | 0.418 | 0.287 |
| Electricity | 96 | **0.125** | **0.216** | 0.126 | 0.217 | 0.120 | 0.222 | 0.139 | 0.237 | 0.134 | 0.230 | 0.143 | 0.247 | 0.158 | 0.266 | 0.140 | 0.237 |
| | 192 | **0.143** | **0.232** | 0.143 | 0.233 | 0.147 | 0.242 | 0.154 | 0.250 | 0.154 | 0.250 | 0.158 | 0.260 | 0.175 | 0.287 | 0.154 | 0.251 |
| | 336 | 0.160 | 0.252 | **0.159** | **0.250** | 0.165 | 0.260 | 0.170 | 0.268 | 0.169 | 0.265 | 0.168 | 0.267 | 0.184 | 0.296 | 0.169 | 0.268 |
| | 720 | **0.186** | **0.277** | 0.189 | 0.279 | 0.199 | 0.289 | 0.212 | 0.304 | 0.194 | 0.288 | 0.214 | 0.307 | 0.200 | 0.310 | 0.204 | 0.301 |
| | Avg | **0.153** | **0.244** | 0.154 | 0.245 | 0.160 | 0.253 | 0.169 | 0.265 | 0.163 | 0.258 | 0.171 | 0.270 | 0.179 | 0.290 | 0.167 | 0.264 |
| Exchange | 96 | **0.078** | **0.196** | 0.080 | 0.197 | 0.083 | 0.200 | 0.082 | 0.199 | 0.086 | 0.205 | 0.079 | 0.200 | 0.079 | 0.203 | 0.080 | 0.202 |
| | 192 | 0.168 | 0.290 | 0.172 | 0.293 | 0.172 | 0.294 | 0.173 | 0.295 | 0.177 | 0.299 | **0.158** | **0.289** | 0.160 | 0.301 | 0.182 | 0.321 |
| | 336 | 0.311 | 0.402 | 0.336 | 0.418 | 0.323 | 0.411 | 0.317 | 0.406 | 0.331 | 0.417 | **0.297** | **0.399** | 0.300 | 0.403 | 0.327 | 0.434 |
| | 720 | 0.755 | 0.649 | 0.855 | 0.696 | 0.820 | 0.682 | 0.825 | 0.684 | 0.846 | 0.693 | 0.751 | 0.650 | 0.745 | 0.665 | **0.578** | **0.605** |
| | Avg | 0.328 | **0.384** | 0.361 | 0.401 | 0.350 | 0.397 | 0.349 | 0.396 | 0.360 | 0.404 | 0.322 | 0.385 | 0.321 | 0.393 | **0.292** | 0.391 |
| Solar | 96 | **0.170** | **0.195** | 0.176 | 0.198 | 0.181 | 0.247 | 0.208 | 0.255 | 0.190 | 0.244 | 0.170 | 0.234 | 0.190 | 0.250 | 0.199 | 0.265 |
| | 192 | **0.180** | **0.208** | 0.190 | 0.209 | 0.200 | 0.259 | 0.229 | 0.267 | 0.193 | 0.257 | 0.204 | 0.302 | 0.226 | 0.284 | 0.220 | 0.282 |
| | 336 | **0.185** | **0.214** | 0.195 | 0.216 | 0.208 | 0.269 | 0.241 | 0.273 | 0.203 | 0.266 | 0.212 | 0.293 | 0.259 | 0.308 | 0.234 | 0.295 |
| | 720 | **0.198** | 0.244 | 0.206 | **0.223** | 0.212 | 0.275 | 0.248 | 0.277 | 0.223 | 0.281 | 0.215 | 0.307 | 0.341 | 0.365 | 0.243 | 0.301 |
| | Avg | **0.183** | 0.215 | 0.192 | **0.212** | 0.200 | 0.263 | 0.232 | 0.268 | 0.202 | 0.262 | 0.200 | 0.284 | 0.254 | 0.302 | 0.224 | 0.286 |
| ILI | 24 | **1.640** | 0.768 | 1.642 | 0.771 | 1.801 | 0.874 | 2.182 | 1.002 | 1.783 | 0.846 | 1.932 | 0.872 | 2.279 | 1.020 | 2.208 | 1.031 |
| | 36 | **1.594** | 0.771 | 1.647 | 0.773 | 1.743 | 0.867 | 2.241 | 1.029 | 1.746 | 0.860 | 1.869 | 0.866 | 2.451 | 1.085 | 2.032 | 0.981 |
| | 48 | **1.516** | 0.780 | 1.601 | 0.773 | 1.843 | 0.926 | 2.272 | 1.036 | 1.716 | 0.898 | 1.891 | 0.883 | 2.440 | 1.077 | 2.209 | 1.063 |
| | 60 | 1.583 | 0.834 | 1.643 | **0.814** | 1.845 | 0.925 | 2.642 | 1.142 | 2.183 | 0.963 | 1.914 | 0.896 | 2.303 | 1.012 | 2.292 | 1.086 |
| | Avg | **1.583** | 0.788 | 1.633 | **0.783** | 1.808 | 0.898 | 2.334 | 1.052 | 1.857 | 0.892 | 1.902 | 0.879 | 2.368 | 1.049 | 2.185 | 1.040 |
| $1^{st}$ Count | | 71 | | 16 | | 2 | | 4 | | 0 | | 3 | | 1 | | 3 | |

Table 8: Fair long-term forecasting results under hyperparameter searching without the "drop-last" trick. The best model is **bold**, and the second best is underlined. Count is the number of the best results.

## D.4 Use of LLMs

During the preparation of this manuscript, we used the OpenAI ChatGPT (GPT-5) large language model as an assistant for language refinement, grammar correction, and style improvement. The model was also employed for suggesting alternative phrasings and generating draft outlines of certain sections, which were subsequently reviewed, verified, and substantially revised by the authors. All technical content, experiments, analyses, and conclusions presented in this paper were conceived, implemented, and validated solely by the authors. The authors take full responsibility for the accuracy and integrity of the manuscript's content.

