# OpenReview forum: "Dual-MoE: Learning Time and Channel Dependencies via Dual Mixture-of-Experts for Time Series Forecasting"
_ICLR.cc/2026/Conference — Submitted to ICLR 2026_

### Official Review · Reviewer_FnY9 · 2025-10-28

**Soundness:** 4
**Presentation:** 3
**Contribution:** 4
**Rating:** 8
**Confidence:** 5

**Summary:**

This paper proposes Dual-MoE, a dual-path mixture-of-experts framework for multivariate time series forecasting that jointly models temporal distribution shifts and noisy channel dependencies. The model consists of two complementary components: the Temporal Fusion MoE and the Channel Fusion MoE. Additionally, a Mask Loss mechanism is introduced to enhance robustness. Extensive experiments conducted on ten real-world datasets demonstrate that the proposed method outperforms existing approaches.

**Strengths:**

**S1.** The paper presents a unified and interpretable framework that effectively addresses the challenges of temporal distribution shift and noisy channel dependencies, with clear motivation and a sound design.

**S2.** Strong methodological contribution through the dual-path MoE design that jointly addresses temporal distribution shift and inter-variable dependency — a long-standing and insufficiently studied issue in time series modeling.

**S3.** Comprehensive experiments across ten datasets, with ablation and sensitivity studies verifying module contributions and showing robustness under noisy or high-entropy conditions.

**S4.** The paper is well-structured, with clear writing and informative visualizations that aid understanding.

**Weaknesses:**

**W1.** The paper introduces a multi-scale lookback time-imaging mechanism, but the process for determining these scales remains unclear. The study currently relies on manually fixed configurations, without discussing whether a systematic or learnable strategy could automatically identify optimal lookback ranges for different datasets.

**W2.** The Mask Loss filters noisy channels using a quantile threshold. How is this threshold selected, and how sensitive is the model’s performance to this hyperparameter?

**W3.** The interpretability of the learned probability matrices in the Channel Fusion MoE is insufficiently discussed. It would strengthen the paper to include visual or quantitative analyses linking these learned relationships to known inter-variable dependencies, thereby substantiating the interpretability claims.

**Questions:**

See **W1-W3**.

---

### Official Review · Reviewer_CLRo · 2025-10-30

**Soundness:** 1
**Presentation:** 2
**Contribution:** 2
**Rating:** 2
**Confidence:** 4

**Summary:**

This paper proposes Dual-MoE, a dual mixture-of-experts framework for multivariate time series forecasting. It integrates a Temporal-fusion MoE for adaptive long–short term pattern modeling and a Channel-fusion MoE for inter-variable dependency learning. A quantile-based Mask Loss enhances robustness to noise. The framework is modular, flexible, and demonstrates consistent improvements across real-world datasets.

**Strengths:**

1. This paper proposes Dual-MoE, a dual mixture-of-experts framework designed to jointly model temporal dynamics and inter-variable dependencies in multivariate time-series forecasting.
2. By integrating a Temporal-fusion MoE that captures both short-term fluctuations and long-term trend dynamics using Exponential Moving Average (EMA), and a Channel-fusion MoE that adaptively models inter-variable dependencies, the framework effectively handles temporal heterogeneity and dynamic correlations.
3. Extensive experiments across diverse real-world datasets demonstrate that Dual-MoE consistently outperforms state-of-the-art baselines, showing superior robustness and forecasting accuracy under varying noise levels and temporal distribution shifts.

**Weaknesses:**

1. Code Implementation-Paper Description Inconsistency
a. The code provided in the Supplementary Material does not implement the distance computation in Equation (2), indicating a discrepancy between the mathematical description and actual implementation.
b. The parameter \delta, described as a threshold for filtering noisy dependencies, is not used in the provided implementation. The paper does not clarify why it was omitted or how this affects performance.
2. Terminological and Conceptual Inaccuracy
a. The term "frequency-aware" used for the Channel-fusion MoE is inconsistent with the implementation, which operates on time-domain patch embeddings without any frequency-domain transformation.
b. Line 196 mentions "Top-K expert selection based on mutual information maximization criteria", but no such mechanism is described or implemented, suggesting either an omission or inconsistency in reporting.
3. Ambiguity and Lack of Empirical Support
a. Figure 3 lacks labels for each time series, and the terms "local fluctuations" and "global trends" are used without clear quantitative definitions.
b. The argument in line 149—that short lookback windows capture local fluctuations while long windows preserve global trends—is supported by only a single example, without dataset-level validation.
c. The ablation study includes EMA removal but not a comparison with Simple Moving Average (SMA), which is a key baseline in prior work (e.g., DLinear). This makes it unclear whether improvements stem from EMA or from the proposed design.

**Questions:**

1. The Supplementary Material implementation does not appear to include Equation (2). Was this distance computation intentionally omitted or replaced during experiments?
2. The paper introduces \delta as a threshold parameter but it is unused in the code—could the authors explain this discrepancy?
3. Why is the Channel-fusion MoE described as "frequency-aware" when the implementation is based on time-domain operations?
4. Could the authors provide more details or an explanation for the "Top-K expert selection based on mutual information maximization criteria," which is mentioned in the text but not implemented?
5. Could the authors clarify how "local fluctuations" and "global trends" are formally defined or measured across datasets? Is there a quantitative evaluation that supports the qualitative claim in Figure 3 regarding short vs. long lookback windows?
6. Have the authors compared the proposed EMA-based decomposition with the Simple Moving Average (SMA) approach used in DLinear to isolate the impact of EMA?
7. In Table 5, were model dimensions such as d_model kept consistent when computing MACs to ensure a fair efficiency comparison?

---

### Official Review · Reviewer_5WjE · 2025-11-01

**Soundness:** 2
**Presentation:** 2
**Contribution:** 2
**Rating:** 4
**Confidence:** 4

**Summary:**

This paper introduces Dual-MoE, a novel and highly complex framework for multivariate time series forecasting. The work is motivated by two challenges: 1) Temporal Distribution Shift, where a fixed-length lookback window struggles to balance long-term trends and short-term shocks, and 2) Noisy Channel Dependencies, where models either over-rely on all channel correlations (like Transformers) or ignore them completely (like channel-independent models). The authors solve this challenge with Dual-MoE with two different mixture-of-experts and quantile-based loss function.

**Strengths:**

S1. The preliminary study in Section 3 is good. It clearly demonstrates the non-trivial, U-shaped relationship between lookback window size and performance (Fig 1), providing a solid, data-driven justification for the proposed multi-scale temporal fusion approach.

S2. The model demonstrates consistent and significant SOTA performance across a wide range of benchmarks (Table 1), outperforming a strong and recent set of baselines (iTransformer, PatchTST, DLinear, FITS, etc.). The high "Count" of first-place finishes (14) suggests the model is robust.

S3. The paper is well-written, and easy to follow.

**Weaknesses:**

W1. The methodology section (Sec 4) is exceptionally difficult to follow and contains several contradictions, making the proposed method nearly impossible to reproduce.

> Figure 4(d) shows the "Temporal-fusion MoE" occurring after the main encoder stack, fusing features from a "Multi-Scale Representation." In contrast, Equation 9 describes the T-MoE as a weighted sum of $Linear(Flatten(X_j))$, where $X_j$ appears to be the raw input from different windows before any deep encoding. These two descriptions are mutually exclusive. Does the fusion happen at the input layer or the output layer?

> Section 4.2 mentions "Multi-scale Lookback Time Imaging" and "ID-to-2D reshaping," strongly implying a 2D-convolutional or 2D-attention mechanism (like TimesNet). However, this is never mentioned again. The "Transformer Encoder" is described as standard (Sec 4.2/C.2), which typically uses 1D inputs. This is a critical, unclarified architectural detail.

> The Channel-fusion MoE (Sec 4.3, Eq 2.) is based on a heuristic formula for the probability matrix P. The definitions of $D_{max}$ and the threshold $\delta$ are unclear. The entire formula appears arbitrary and lacks theoretical or strong empirical justification.


W2. The proposed model itself seems a combination of the previous works and marginally improved version of the other works, including M-scale inputs, EMA decomposition, utilization of multiple MoE, and quantile-based loss functions.

> More importantly, the core concepts, especially the MoE with multiple roles [1] and the quantile-based loss functions [2] are already proposed in the previous works but is not cited in this paper.

[1] TESTAM: A Time-Enhanced Spatio-Temporal Attention Model with Mixture of Experts, in ICLR'24

[2] Spatial Mixture-of-Experts, in NeurIPS'22

W3. The efficiency analysis (Table 5) is weak. It compares MACs and inference time against "Pathformer" and "PDF," but not against the SOTA baselines from Table 1, such as the highly efficient DLinear, iTransformer, or FITS. This feels like a cherry-picked comparison. Given the model's complexity, a comprehensive efficiency benchmark against the actual competitors is necessary.

W4. The Mask Loss (Sec 4.5) is a problematic component. It essentially "gives up" on channels the model finds difficult. While this may improve aggregate MSE, it is undesirable in real-world scenarios where all channels must be forecast. Discussion about this concern is also discussed in the previous papers [1,2], but this paper missed it and make the optimization slow.

**Questions:**

Q1. Does the Temporal-fusion MoE (Sec 4.4, Fig 4d, Eq 9) fuse the raw multi-scale inputs (like Eq 9 suggests) or the encoded representations from a deep stack (like Fig 4d suggests)? These are fundamentally different models.

Q2. What is "Time Imaging"? Is the Transformer Encoder (Sec 4.2) a 1D model, or is it a 2D model (like TimesNet) that operates on the reshaped 2D "images"? This is a critical, missing architectural detail.

Q3. Can you provide a full efficiency comparison (MACs, training time, inference time) against the primary SOTA baselines from Table 1, especially DLinear, iTransformer, and PatchTST? The comparison in Table 5 is insufficient.

Q4. Can you provide a better justification for the Mask Loss? How does it affect the performance on the worst-performing (masked) channels? Does it not simply learn to ignore the most difficult, and potentially most critical, signals?

---

### Official Review · Reviewer_h2qw · 2025-11-01

**Soundness:** 2
**Presentation:** 2
**Contribution:** 2
**Rating:** 2
**Confidence:** 4

**Summary:**

The manuscript introduces Dual-MoE, a dual mixture-of-experts (MoE) framework for multivariate time series forecasting (MTSF). The model integrates two complementary modules: (1) a Temporal Fusion MoE that dynamically balances long- and short-term temporal dependencies via exponential moving average (EMA) and multi-scale lookback windows, and (2) a Channel Fusion MoE that models inter-variable relationships using a learnable probability matrix based on frequency-domain similarities. A quantile-based Mask Loss Function further enhances robustness to noisy or unpredictable channels.

**Strengths:**

1. The writing is easy to follow.

**Weaknesses:**

1. **Dispersed and Unfocused Innovation**
   The paper’s innovation appears fragmented. Both *Temporal Distribution Shift* and *Channel Dependencies* are substantial and independent research problems, each worthy of separate, focused exploration. Presenting them together dilutes the novelty and makes the work feel more suitable for a journal article than a conference paper, which typically values concise and insight-driven contributions.

   Furthermore, the paper seems to conflate short-/long-term dependencies with lookback window size. A larger temporal window already contains information about both short- and long-term patterns; thus, the real challenge lies not in window length itself, but in whether the model can *learn to attend to the right historical segments dynamically*. The current motivation does not make this distinction clear.

   On the other hand, *Channel Dependencies* have been extensively studied in spatiotemporal forecasting literature. Although the datasets used in those works differ slightly in semantics, their data structures are identical, and the corresponding methods are largely transferable. This relevant body of work is not sufficiently acknowledged or contrasted, which weakens the originality of this component. Finally, the fourth paragraph of the Introduction is overly generic and lacks a clear connection to the two motivating problems discussed earlier.

---

2. **Overly Engineering-Oriented Method Design**
   The proposed framework introduces many modules, but the motivation and underlying insights for these design choices are insufficiently analyzed. This again fragments the novelty and makes the contribution appear as a collection of engineering add-ons rather than a coherent conceptual advance.

   For example, line 56 states that *“The Temporal Fusion MoE formulates the fusion of long- and short-term temporal patterns as a classification problem.”* However, the paper does not explain *why* this formulation is appropriate or advantageous, nor does it clarify *why* the Exponential Moving Average (EMA) is specifically chosen to track trends and distributional shifts. Without these theoretical or empirical justifications, readers may find it difficult to grasp the key insights that differentiate this work from existing architectures.

---

**Overall**, the paper lacks a clearly articulated motivation, concrete evidence supporting that motivation, and insightful reasoning linking the identified challenges to the proposed design. The methodological innovations are incremental and insufficiently conceptual to meet the originality standard expected for ICLR-level contributions.

**Questions:**

See weakness.

---

### Meta-Review · Area_Chair_7N1G · 2025-12-23

**Summary:**

The reviewers' core concerns include: scattered innovation with insufficient connection to prior work, discrepancies between method descriptions and code implementation, poor reproducibility, inadequate empirical validation, and potential neglect of critical practical issues. The sole positive evaluation acknowledged the framework's consistency and experimental performance but did not address the aforementioned shortcomings.

**Reviewer Concerns:**

**Addressed by Rebuttal**:

None (no rebuttal was provided by the authors).

**Still Outstanding:**

Fragmented innovation, insufficient connection to prior work, and unacknowledged overlaps with existing MoE/quantile loss methods.

Contradictions in methodological descriptions and inconsistencies between theory and code.

Incomplete efficiency comparisons, lack of ablation against key baselines, and inadequate justification for hyperparameters/design choices.

Practical concerns about the Mask Loss ignoring critical channels, unclear hyperparameter selection, and insufficient interpretability of learned dependencies.

**Reviewer Scores:**

Since neither the authors nor the reviewers provided further comments or discussion, I believe their ratings would not change.

---

### Decision · Program_Chairs · 2026-01-26

Reject